# At Home: Place Attachment and Identity in an Italian Refugee Sample

**DOI:** 10.3390/ijerph18168273

**Published:** 2021-08-04

**Authors:** Caterina Nicolais, James Michael Perry, Camilla Modesti, Alessandra Talamo, Giampaolo Nicolais

**Affiliations:** 1Faculty of Medicine and Psychology, Sapienza University of Rome, 00185 Rome, Italy; perry.1650817@studenti.uniroma1.it; 2Department of Social and Developmental Psychology, Faculty of Medicine and Psychology, Sapienza University of Rome, 00185 Rome, Italy; alessandra.talamo@uniroma1.it; 3Department of Dynamic, Clinical and Health Psychology, Faculty of Medicine and Psychology, Sapienza University of Rome, 00185 Rome, Italy; giampaolo.nicolais@uniroma1.it

**Keywords:** place attachment, refugees, home, identity

## Abstract

The central tenet of Place Attachment theory states that an individual has an inborn predisposition to form strong bonds with places as well as with people. Our qualitative study applies this theory to understand how, despite loss and adversity, refugees are able to reconstruct a sense of identity, community, and “home”. Participants included 15 forcibly displaced people from different countries of origin. Semistructured interviews explored factors that facilitate participants’ integration in a new context and the impact of this context on their sense of identity. Data were analysed using Consensual Qualitative Research Methodology to identify recurrent themes and their frequencies within interview transcripts. Within the relational dimensions of *place attachment*, *affiliation*, and *seeking help from others*, the study explores the factors that facilitate the integration of refugees in a new context and the impact of this context on their sense of identity, identifying recurrent themes and their frequencies within interview transcripts. The most frequent resulting themes were (a) a sense of identity and (b) expectations toward the resettlement country. Additional, though less frequent, themes included: (c) sense of belonging, (d) community integration, (e) trust, (f) opportunity seizing, (g) being a point of reference for others, (h) sense of community, (i) positive memories, (j) refusal. These results begin to describe the ways by which Place Attachment, toward both birth and resettlement countries, contributes to a restructured identity and sense of “feeling at home” for refugees.

## 1. Introduction

A refugee is a person forced to leave his or her country due to armed conflict or persecution for reasons of race, religion, nationality, political views, or belonging to a particular social group. Unlike a migrant, a refugee has no choice and cannot return due to fear of persecution or death [1].

The flow of migrants to Europe, particularly in Italy, has increased significantly in the last decade. Italy saw more than 150,000 people arrive at its borders and coasts, most of the time involved in emergency sea rescue with countless victims. Stress on existing reception systems and integration has been significant [2,3], and the links between migration and increased risk of developing psychiatric disorders are well documented [4,5].

Each of these individuals had to leave their homes and start over, reassessing identity in a new cultural context. Building a new home in a foreign country is a process that has much to do with attachment, in that the quest for security and rootedness always implies the activation of that motivational system [6]. Literature does not present research or studies about place attachment in refugees, though the many and evolving definitions of this construct often refer to migrants [7,8,9,10,11,12,13,14,15,16,17,18].

This qualitative study explored the relational dimension of place attachment, affiliation, and seeking help from others in the framework of attachment theory within a sample of association founders or leaders in Italy, each themselves individuals with refugee status or subsidiary protection. For the sake of clarity and ease of comprehension, all forcibly displaced individuals with international protection status will be referred to as refugees.

### 1.1. Theoretical Context

#### 1.1.1. Attachment and Place Attachment

According to Attachment Theory [19], the development of a pattern of secure attachment is both a protective factor, as far as security is concerned, and the precondition for the exploration of the environment. In agreement with this theory, so-called “secure base” behaviour is part of our genetic heritage. This is not limited to the activation of relationships with sensitive and responsive attachment figures who meet our needs and to whom we can turn as a safe haven but can also be extended to the way we relate to the environment and establish effective bonds to places we live [11]. Indeed, according to Bowlby, “there is a marked tendency for humans, like animals of other species, to remain in a particular and familiar locale and in the company of particular and familiar people” [19] (p. 147). Within this perspective, place attachment [7,8,9,10,11,12,13,14,15,16,17,18] is a process peculiar to the attachment motivational system, where we interact with our physical environment to gain a sense of protection and comfort.

Separation, loss, and the search for security are not only the main themes of Attachment Theory but the lived experiences of migrants. The study of these themes in migrant populations sheds light on some of the processes of adaptation in the face of severe adversity while describing a “phenomenology” of place attachment. As a result, a more nuanced understanding of the different phases of migration, pre-migration, transit, adjustment, and adaptation, [20] is possible in dialogue with the psychological variables at play in place attachments [21].

During the years, the definition of Place attachment has been imprecise and unclear. This unclarity is due to the difficulty in measuring this construct and due to differences in cultural settings.

As a psychological construct, place attachment was first introduced in the early 1960s. Fried [7] was the first to describe the negative consequences of a forced dislocation that causes a reaction similar to what is commonly observed when a romantic relationship ends, with sadness and emptiness as leading forces of mourning-like dynamics.

Tuan [8,9] regards place attachment as a fundamental human need. Focusing on its affective and spatial aspects he introduces the concept of *Topophilia*. This dimension occurs when the emotional meaning attributed to the geographical space can transform it into a “place”. Along this line, he distinguishes between rootedness and a sense of place. The former is defined as an unconscious state of deep familiarity with a place that implies a lasting residence. The second is referred to as a conscious force of creating and preserving places through words, actions, and the construction of artifacts.

Stokols & Shumaker [10] theorize that place attachment is the result of cognitive representations and emotional bonds that people have with places. They suggest that it is characterized by two dimensions: spatial identity and spatial dependence. The latter refers to how much the place satisfies the needs of the individual, while the former indicates an emotional bond that is structured when the place takes on a symbolic meaning and becomes part of the person’s identity. Attachment as a positive emotional bond or as an association between individuals and their residential environment can operate on both an individual and group level and declines when it no longer satisfies the needs of individuals. As is true of interpersonal attachment, multiple attachments to places are possible. The persistence of attachment to a specific place has been called by the author’s geographic place dependence.

Considering the negative effects for those not securely attached to a place, Hummon [11] introduces five types of place attachment. Each type describes a person’s cognitive processes related to place: everyday rootedness, a place attachment taken for granted; ideological attachment, a more reflective and active attitude, typical of people who have deliberately chosen to live in a certain place; place alienation, identified in those who have a negative attitude towards a place of residence and present the desire to move elsewhere; relativity, in which people appreciate the positive aspects of where they live, but may also live elsewhere; finally, placelessness, present in people who do not consider the spatial dimension as important or apply special meaning to places.

In the same vein, Giuliani [12] states that place attachment is a positive or negative emotional bond with a place that can be related to our current or past experience. For this author, every relationship is influenced by place attachment, given that there is no mutual affinity, community, fraternity, or feeling of diversity, aversion, resettlementality, which is not somehow influenced by contextual variables.

Rollero and De Piccoli [13] argue that attachment to significant places depends on the mutual relationship between behaviours and experiences. They refer to different sizes of places that people are attached to. The first concerns the affective role and functions such as emotional attachment to the physical and natural elements of a place, an element that therefore describes what the person is attached to. The second concerns the “life worlds” and the natural attitudes of a place. This dimension is called process dimension and describes how attachment manifests. The third, the person dimension, includes a cognitive function and concerns the unique character of a place that enables the person to create a place identity.

According to the perspective put forward by Scannell and Gifford [14], place attachment can be framed according to the personal dimension (individual or collective, concerning the actor), according to the spatial dimension (physical or social, concerning the object of the attachment) and finally according to the procedural dimension (affective, cognitive, and behavioural, concerning the psychological process). Among factors connecting interpersonal attachment and place attachment is the fact that both represent a protective factor in the face of stressful life events. It is therefore assumed that a place can act as a substitute for an attachment figure.

Adapting an earlier questionnaire by Bartholomew and Harowitz on interpersonal attachment [20], McBain [16] demonstrated a positive correlation between interhuman and Place Attachment. Individuals with secure attachments felt more secure about their current home and reported fewer avoidant and anxious feelings. Place Attachment, like interpersonal attachment, seems to be stable over time, although influenced by factors such as age and socioeconomic status.

Affectively relevant places can be a refuge where a person can withdraw from the negative events of his life, thinking about solutions, and ultimately finding emotional relief. This can be of relevance for individuals and marginalized groups faced with numerous stressful events in everyday life [21]. This is similar to what is found in interpersonal attachment, but according to Scannell and Gifford [15], a place can also operate as a safe haven in the absence of a human caregiver. It is noteworthy that for the authors “home” may not always coincide with the secure base. In some cases, it is the place where abuses or violence may occur, so that the individual may look for a secure base in different and safer environments.

People appear to show a common disposition to establish and maintain an attachment to places. Scannell and Gifford [15] do not define place attachment as universal as Bowlby [19] did with respect to interpersonal relationships. In both cases, however, attachment is not exclusively directed towards a single place or person but can also be established towards different places and different people, although the authors emphasize that the hierarchy of places and their relationship with interpersonal attachment figures is an area not yet investigated [15].

#### 1.1.2. Place Attachment in Migration Studies

In literature, there are few studies investigating the issue of attachment to one’s own country, as well as in the resettlement country, in migrants [17,18,22]. There are no studies that observe which factors determine the relationship between the place attachment pattern before migration and the pattern established in the resettlement country. Besides, in literature, there are no studies investigating the issue of attachment to one’s own country, as well as in the resettlement country, in refugees.

The refugee population is forced to leave their homeland due to persecutions and violence [1]. Therefore, the relationship with their birth country and with their community is suddenly interrupted. This condition is radically different from migrant conditions, where people actively decide to migrate to a new country to find better living conditions. Thus, it is important to understand how a new place attachment can be determined despite the negative life event in a successful migrant population like ours.

Nevertheless, as the literature about place attachment shows, it seems particularly meaningful to study place attachment in the migration population as it is strictly connected with the adapting process to the resettlement country. Indeed, this connection often brings about a fair integration. Even before their arrival, newcomers are inserted into communities: the resettlement and the origin ones but also in religious communities and so on. The relationships that newcomers tie with these communities and the connections among them will shape their social integration path within the resettlement country. In order to access the resources produced by group dynamics, individuals need to recognize themselves and to be recognized as effective members of specific groups. This issue calls into question the topic of Identity. Therefore, in order to explore how group dynamics lead to social integration, the concept of Identity needs to be addressed and explicated.

Place attachment and sense of belonging to the local community bring the newcomer to be a local citizen. In this way, migration is not temporary or a way to improve living conditions anymore, but it becomes a life project.

This process seems to be strictly connected with the link with the birth country. Indeed, in exploring place attachment in migrant populations, Gustafson suggests that it is important to take into consideration both the bond with the country of origin and the resettlement country [17]. Migrants generally maintain links with their countries of origin thus strengthening their attachment to the place of origin, while concurrently establishing a peculiar kind of place attachment in the resettlement country.

According to Trąbka [18], the link with a new place of residence is a dynamic and non-static process. Different aspects of place attachment often coexist and can gradually emerge in the process of adapting to a new context. The author identifies different dimensions of Place Attachment. Firstly, place dependence is a form of relative attachment, often following reflection on the positive and negative aspects of life in a certain place. In this case, a person is attached to a place for a particular reason and, once those needs change, another migration can take place. Place discovered describes a situation in which the person moves beyond “passive” residency to create a deeper bond with the city. This process is self-referential and involves agency, commitment, and participation. The third dimension is called place identity, a process requiring at least three years, in which the heart of the person is present, and the place begins to play a fundamental role in their sense of self. Finally, the place’s inherited dimension is a question of habit, a strong bond established in a lifelong place of residence. In migrants, this occurs in cases where the person emigrated at a young age and lived in the place for at least ten years. The study concludes that these aspects can also coexist. Some of them emerge quickly (the first and the second), while the other two need more time to evolve. The manifestation and interplay of these dimensions depend on different socio-demographic factors such as age, family situation, social and cultural capital. Furthermore, some migrants may not encounter any of the dimensions and may manifest the alienation described by Hummon [11].

Van Ecke [22] presents a different point of view, theorizing that the migrant population is more likely to have insecure attachment than the non-migrant population. Since attachment theory distinguishes between separation and loss, even for the migratory phenomenon we should distinguish between the beginning and subsequent phases of immigration. The latter adds a sense of isolation and loss to the initial discomfort. Therefore, the most critical phases are not the first ones, as more often there are a series of smaller events that add up later and that come to constitute a real traumatic event. Moreover, over time, the migrant no longer can participate in positive events with family in the country of origin, nor to assist or support them. The migrant’s representation of place attachment could therefore suffer from the continuous losses accumulated over the years.

Perez-Foster [21] uses attachment theory to give meaning to the responses of immigrants during the four typical stages of migration: pre-migration, transition, settlement, and adaptation.

As for the pre-migration phase, according to the studies by Boneva and Frieze [23], those who decide to emigrate have greater fulfilment and power motivation, are more work-oriented, and are less family-focused. The aforementioned characteristics, as van Ecke [22] explains, are also found in people who have an insecure, distancing attachment pattern. As for the adjustment phase, those who have a distancing attachment initially may adapt more quickly, while those who have a worrying or secure attachment have greater difficulties. In the resettlement phase, the author introduces the concept of acculturative stress, [24] which indicates the loss of familiar elements, sounds, and faces, together with the sense of not knowing where one belongs, how one connects to others, and how to seek support. The author refers to changes that groups and individuals face when they meet a different culture. He emphasizes the emotional burden that this process entails: it is hypothesized that, during this event, high levels of stress can lead to depression and suicidal ideation. Some psychological factors that can contribute to increasing this risk or to mitigate it are the social support given by the new community, the new supportive family networks, the socio-economic status, the work and educational changes, as well as some pre-migratory variables of personal functioning such as self-esteem and coping skills, knowledge of the new language and the new culture, control, and the decision to migrate (voluntary or not). In the last phase of adjustment and adaptation, the author hypothesizes that socio-economic security contributes to determining a secure attachment pattern towards the resettlement country, even if lower mental health seems to be related to the major separation from and permanent loss of the attachment figures.

Cited articles, despite their heterogeneity, lead us to hypothesize that a migrant, as also a refugee, is able to find a new home and identity in the resettlement country, while continuing to feel a sense of belonging to the birth country and former identity.

According to the literature, we believe that place attachment in the refugee population is influenced by the attachment towards the birth country and that it can fully develop thanks to the positive integration to the resettlement community.

Our study aims to explore which factors and dynamics underlie the development of new place attachment and identity in the refugee population.

## 2. Materials and Methods

### 2.1. Purpose of This Study

This research aims to understand in which way place attachment develops regarding the resettlement country and in what ways does it relate to birth country place attachment in refugees who had positive integration social experiences. Indeed, participants were selected as members of social enterprises with migratory background or rather social enterprises founded by former refugees who were granted with the Italian citizenship and were committed toward newcomers’ social integration. As such, those people used their own first-hand experience in social integration in order to ease the integration path of those who came after them. For this very reason participants were selected exactly for their experience of positive integration [25,26].

This research was conceived as an exploratory study and, as such, proposes two research questions in the place of hypotheses:RQ1: Which factors determine the creation of a new “home”?RQ2: How does the new refugee identity develop and how it relates to established relationships in the resettlement country, in particular through affiliation and seeking help from others.

### 2.2. Participants and Procedure

The study was composed of fifteen participants: ten of them were refugees (R), two had subsidiary protection (SP) and finally, three of them were asylum seekers (AS). Fourteen of them were male and one was female. The average age was 33 and the average time lived in Italy was 10 years. Participants represented 10 countries in Africa and South Asia, with the largest groups from Afghanistan and Somalia.

Participants were involved on a voluntary basis without any payment or reward. They signed the informed consent that described the research objectives and procedures, the collection and processing of data-related procedures in compliance with ethical and privacy regulations.

The refugees represented a group of refugees who had positive integration experiences. Indeed, they came from five different associations. Each participant was a founder or a leader of a registered community association in Italy, constituted to support multi-ethnic groups of migrants through information sharing, advocacy, material support, cultural integration in Italian society, or job placement. Participant information is summarized in Table 1.

### 2.3. Measures

The data was collected from January 2019 to April 2020 and the Ethics Committee of the Department of Social and Development Psychology issued the Declaration of Ethics Approval.

Researchers began by identifying associations in Italy with a migratory background that had invested in the social capital of members and beneficiaries. The data collection procedure took place by contacting the association leadership, describing the research procedure and objectives, and obtaining consent for data collection. The interviews were conducted face to face, lasted an average of 60 min and were audio recorded. The audio records were encoded with alphanumeric codes (for example GP001) to preserve the anonymity of the interviewees. The live interviews were conducted in the association’s headquarters or in quiet neighbouring places where it was possible to conduct the interview in private. Given the geographical distance, two interviews were conducted by video call. All the collected material was subsequently transcribed anonymously and explored using the Consensual Qualitative Research (CQR) [27].

The tool used for data collection was a narrative interview with an open structure, which aimed to collect information about specific events that represent the thread of the narrative. The interview was built based on Atkinson’s model [28], as it captures the interviewees’ point of view and the systems of meaning they refer to interpret specific events. The open-ended questions allowed participants to describe their experiences in reference to predefined topics. The thematic areas explored through the interviews were: the reasons that motivated migration, the process of social integration in the resettlement country, and the establishment of the association and its objectives, activities, and beneficiaries.

### 2.4. Data Analysis

Using the Consensual Qualitative Research (CQR) method [27], domains, categories and subcategories were identified in each interview in relation to the guiding research questions. This procedure identifies and quantifies the most frequent themes in the interviews. The frequency of themes was described using Elliot’s [29] classifications: general, a category that applies to all cases; typical, if it is applied to half or more of the cases; and variant, if it is applicable to two or three cases, or to less than half of the sample.

Firstly, *consensus* is a necessary condition for Consensual Qualitative Research (CQR), and it was formed by sharing perspectives formed through regular research meetings in which the team shared perspectives and debated initial reflections and categorizations of the data. Then, we proceeded to the description of the research objectives that could however be subject to change during the data collection journey. At this point, we proceeded with the choice of the instrument to be used for data collection. Subsequently, the information obtained through the interview was audio recorded and then transcribed. According to the CQR guidelines, the number of participants is not too wide, between eight and fifteen participants, in such a way as to have an overview, but not too broad and dispersive of the phenomenon. Once the preliminary stages have been completed, we proceed with the analysis and control of the data collected. The purpose of analysing the data is to accomplish a clear explanation of the experiences told by the interviewees [30].

The first phase of CQR consisted of dividing answers into domains, which are representative and descriptive themes of the observed phenomenon that are common to all interviews. These are sections created through what emerges from the story of the participants. In addition, they have to deal with every different aspect of what is being investigated and being mutually exclusive. The next phase involved the construction of the central ideas through the “cleaning” of the text assigned to each domain, or the transformation of the narration into clear and non-repetitive language. Categories were built so that they can group the units of meaning related to the various domains thus allowing to find similarities between the narratives. Once the units of meaning related to the different domains have been grouped, they were assigned a category. The last step of the Consensual Qualitative Research was frequency analysis, which explored the representativeness of categories by counting cases within each category. The aim was to compare what emerged during the narratives of the subjects. To do this, the cases that fall into that category were added together. The goal of this phase was to perform the frequency analysis to check if the single category was representative of the entire sample.

## 3. Results

As Table 2 shows, 10 domains were identified. They contain the macro-themes most descriptive of the functioning of the refugees in our sample. The two main domains were: (a) sense of identity and (b) expectations toward the resettlement country. Additional domains, though less frequent, included: (c) sense of belonging, (d) community integration, (e) trust, (f) opportunity seizing, (g) being a point of reference for others, (h) sense of community, (i) positive memories, and (j) refusal.

Within each domain, individual categories and subcategories were identified to describe the refugees of that domain.

Frequency analysis revealed various domains common across all participants, while others described smaller subsets.

Following the steps of consensual qualitative analysis (CQR) it was finally possible to formulate a reflection on the collected material. Starting from the exploration of the individual domains, we described the categories and subcategories associated with them to give an overview of the place attachment of the sample and their identity.

### 3.1. Categories

#### 3.1.1. Sense of Identity

“Sense of identity” is one of the domains that emerges most clearly. It has been defined as one’s own perception of belonging to a national group. In particular, it was manifested towards the birth country, the resettlement country, or both of them. It has been divided into three categories.

##### Acquiring the Identity of the Resettlement Country

This category refers to those refugees who completely identify with the resettlement country. It includes two sub-categories: “desiring to learn the language,” defined as general as it emerges in almost all the interviews; and “adapting to the culture”, typical, present in more than half of the sample. In fact, refugees stated that one of the best ways to feel part of a new place was to learn its language and integrate culturally. In most of the interviews, refugees described how language had been a crucial tool for good integration. For example, referring to the sub-category “desiring to learn the language”, one of them reported:

(…) and that person who worked in the dormitory advised me to go to school: “Instead of spending time outside, why don’t you learn the language?”, Which was a very positive indication for me (…).

These factors were perceived as the only way to forge relationships with local people:

(…) I tell you sincerely that I feel more Italian than Pakistani now, and there are reasons why (…) Turin was the first city that opened all its doors to me, that gave me all the possibilities, what I dreamed of in Pakistan (…).

##### Maintaining the Identity of the Birth Country

This category includes the refugees who, despite the dislocation, still identify with the birth country. It has two sub-categories that concern “cultural and linguistic adaptation”, and “difficulty in integrating”. However, these two sub-categories appeared to be variant. In fact, our sample, by its very nature composed of members or presidents of associations, includes people who, once successfully integrated, decided to share their experience with others:

I have always said that we are Somalis and therefore we belong to Somalia.

##### Having an Identity with Both Countries

Finally, this category is one of the most frequent categories (general). It refers to refugees who developed an identity both with their birth country and with the resettlement country:

(…) I do all this together with the Italians (…) if someone wants to remove me by calling me a Somali citizen, I say that I am both a Somali citizen, but I am also an Italian citizen (…)

The high frequency of transcriptions falling into this category means that most of the sample managed to integrate positively and build an identity between the two countries. In fact, people involved in our research are all at the head of associations or its members, therefore they are people who “made it” and who want to dedicate themselves to help other migrants.

#### 3.1.2. Expectations toward the Resettlement Country

In this domain, all the categories and subcategories were identified in almost all the participants (general). In fact, even at different times during the interview, everyone talked about their expectations towards the resettlement country before embarking on the journey. This domain was divided into two categories.

##### Derived from Difficulties in the Birth Country

This category embodies the transcriptions where the refugee talked about the decision to emigrate as a consequence of difficulties in the birth country. It has two sub-categories: “few opportunities” and “difficult conditions”. The latter sub-category, due to the nature of the refugee condition, has been frequently encountered:

(…) Yes, I was not well in Pakistan, Iran, Turkey, because I always lived in the Pakistani community which was too dangerous, especially in Iran and Turkey; Pakistani boys are dangerous and, as soon as I got out of Turkey, I said to myself: “Okay, I’m looking for a place where I can live my life and that’s it”.

##### Derived from a Sense of Responsibility

This category refers to the decision to emigrate as a consequence of a sense of responsibility towards one’s family or country. In fact, a refugee is someone who emigrated from his country because of difficult living conditions such as, for example, wars or persecutions. In addition, there may be a motivation to leave to guarantee your family a better future or a better present, but also the goal of making a personal contribution to the resettlement country.

The sub-category “towards the resettlement country” was classified as general, since it is present in almost all the interviews:

(…) I felt that I was also doing something for the city I belong to with that association, and I hope that we will be able to do many things in Turin, in Italy, and in our birth countries (…)

as also the sub-category “to the birth country”:

(…) at home I call some friends to make a contribution, some money, so then we can take a house for the children, some clothes when the holidays arrive, when there is something, we give it away, even now: I have already sent 200 euros to some friends near where I lived before, in my country (…)

#### 3.1.3. Sense of Belonging

This domain presented only one category, feeling at home, which was present in about two-thirds of the sample (typical).

In our study, the word “home” refers to a place to which a refugee has developed the deepest attachment. The role and meaning of places such as “Home” have received considerable attention in psychology [31,32]. The attention of researchers focused especially on the subjective, phenomenological experience of home, beyond physical and material aspects of residential environments.

Participants have often mentioned this word, so we decided to entitle this category as “home”, even if we use it as a synonym of place attachment.

Participants reflected on the successful creation of a new home in their resettlement country:

Now I’m from Turin, so for me, Turin is the best, even though I’ve travelled the world, and in any case, it’s always home (…).

Many refugees talk about their “home” as a place where, today, they feel they belong. It is a place where they feel well integrated, where people are kind and where they feel part of a family. In these people, “home” is not just the place where they are no longer persecuted. Many of the refugees we interviewed stated that, even if they could go to many other places, in Europe or in the world, they choose not to do it because they now feel they belong to their current city or country:

(…) I now also have the opportunity to go to France, to go to Germany, but I want to integrate into Italian society, because I was lucky enough to be in Italy and I had things to eat […] in Italy I saw that if you don’t know a guy an Italian, so you think he’s bad, but you have to know him to know that he’s not like that inside. He does not trust you at first but then, if you go further, he will trust you […] he will trust you forever, and this is a good thing, yes, in other countries it is more difficult […] yes, I have already decided to stay here, already decided, yes (…).

#### 3.1.4. Community Integration

This domain refers to the refugee’s success in integrating into the community, which refers to job, social and personal satisfaction. It has three categories.

##### In the Italian Community

Even if the interviewed refugees managed to build a new life in Italy, especially in the initial stages of integration, this integration has often proved complex:

(…) there are always inputs that tell me: “Remember that you are a refugee anyway, remember that some things you cannot do, remember that you do not have the freedom of movement, remember that you cannot go to some places” (…).

##### In the Community of Foreigners

Very often refugees came from fragmented cultures and societies composed of many different ethnicities and religious or cultural affiliations. These contexts lead people to isolate themselves in their group and to develop discriminatory attitudes towards others, attitudes which some participants brought with them to Italy. Instead, in Italy, the cultural substrate is more cohesive, and this was noted with pleasure by the participants. While migrants often complained about the reception system and local authorities, analysis of the interviews did not reveal frustrations with private citizens:

(…) Even when I applied for the residence permit, the staff at the reception centre sent me a message saying that I had had a negative answer, but it was not true. They sent me a voice message on WhatsApp, but I called Rome to find out if it was true or not, and they told me it wasn’t true, that I had a positive response, that I had had a residence permit for five years, but they had sent me the message to say that I had had it negative, even if this was not true (…).

Also, this integration has often proved complex:

(…) I had a bit of difficulty with regards to some people I met here, right here in Italy, who come from my country and say bad things about my ethnicity (…).

##### Non-Inclusion

Another category that emerged from the interviews is that of “non-inclusion”, which is the referred missing inclusion in the community. Although it turned out to be infrequent and was thus classified as typical, both in the sub-category of the “resettlement community”:

The important thing is not to ask for asylum, but to live. In the end, if I come from there, I end up here in the countryside (…) why do I have to seek asylum? (…) and these are things I really say are a bit difficult to even get along with and this especially you know with whom? With those who think that it is on the part of migrants the problem is there, not of who everything is concentrated there and therefore it is a continuous struggle (…) lately I wrote an article, I wrote for the post (…) lately I have also sent out some slightly provocative articles that they have always put aside, because people expect you to always say thank you, but I say: “Like hell, thank you, first of all I don’t even want to be here for three years, I lived like hell, I was obliged by the Dublin Regulation to stay. My friends who did not apply for asylum here today have nationality (…).

and in that of the “community of foreigners”:

(…) because to be honest I lost so much work, so many job opportunities for my ideas, now any organization does not want me (…).

#### 3.1.5. Trust

Trust of others is essential when facing a journey into the unknown. In fact, the most integrated people were also those who had established more relationships with others and were able to rely on them. This domain was divided into three categories.

##### Seeking Help from Others

This category highlighted the support that these refugees have received. This has allowed them not only to migrate, but also to study and dedicate themselves to others. This category has two sub-categories: “in the resettlement place” and “in the birthplace”. The latter was classified with the variant frequency, while the former was typical. This could highlight that an important prerequisite to be able to undertake such a trip is knowing how to trust people both before and after the very act of emigrating to a new country, also because this trip was often undertaken alone:

(…) I created this relationship with some educators who were there, so we created this trust between us (…).

##### Self-Confidence

We have identified this category as an important factor for success. In fact, in difficult moments or phases in which relationships with other local people have not yet been established, this dimension is decisive. This category was infrequent (variant), but that is likely since the narrative interview did not specifically explore this theme:

(…) because you know, when you are with your family you always expect something, while when you live alone you go looking for it … let’s say that in the family I am the one who has the motivation that not all boys have … but in the end I said: “No, I want to walk alone”.

##### Difficulty in Trusting

Finally, the last identified category was only observed in one person. Hence, it was classified as a variant, as we have rarely heard a migrant claim that he had difficulty trusting others and himself:

(…) it was difficult to trust even the people close to you … it was also difficult to trust a brother (…).

#### 3.1.6. Opportunity Seizing

The category that we have identified in this domain is that of the cultural exchange, as the interviewee often spoke of the new experiences lived on Italian soil as a way of enriching not only one’s own cultural experience, but also that of the Italian community, a category that had a typical frequency:

(…) With my association we do African things, especially I work with African children. I have learned many things, many dialects, cultures, foods … I have learned many things myself, I have improved my life let’s say, and they too, there is. It was an exchange of culture. I understood that things can be done for the city (…).

In fact, one of the central characteristics of participants is the motivation to build new projects and ideas to allow both cultures not only to “adapt” and “coexist”, but above all to constitute a pluralist society. In fact, they dedicate themselves not only to helping other migrants, but also enriching the relationships between the foreign and Italian communities:

(…) that lady who created this group to unite us, to teach us; some Italians also want to learn drums, but they don’t know how to do it, so with this group they manage to come towards us to learn this material (…).

#### 3.1.7. Being a Point of Reference for Other

All the participants of our study were in some way a point of reference for other citizens, both belonging to the community and to the association. This domain was divided into two categories.

##### For the Birth Community

Nevertheless, the participants of the study are a point of reference not only for the resettlement community, but also for the birth country:

(…) I had this relationship especially with the Pakistani community. I was already a point of reference before founding this association, because already about one hundred and fifty people who were with me out in a park knew me since I was a mediator for them in English. Then they met other people here in Turin, they introduced me and slowly we started doing cricket sports. I was an important person in the Pakistani community and now whatever happens in Turin I know what I have to do, and that’s why they always call me (…).

##### For the Resettlement Community

Most of the refugees we interviewed were leaders or active members of associations with a migratory background. Therefore, we can hypothesize that they were natural leaders and that they presented unique skills and gifts that led them to be a point of reference not only for the work group, but also for the social ones:

The Italian friends I’ve had have always said: “If we ever had a mentality like you, we would go further!”, They always say: “You are patient, you are not an asshole, you are not smart, you are always honest!”. Because sometimes I make small gestures between us, between us friends, when we go for example to dance, or when we go out to dinner … They always say, “You are one that not even Italians, not even we Italian boys can be like you!”. So, some people took me as an example regarding my behaviour and my way of life. I was just an African boy among them, but they did not consider me as an African boy, all they said was: “No, we have to wait for him”. I had good behaviour with them and so in the end they took me as one of them.

#### 3.1.8. Sense of Community

When family, friends, and reference groups are lacking, it becomes essential to rebuild a social group with the people who are in the resettlement country. This domain was divided into two categories.

##### With People from the Birth Country

The “sense of community” of refugees emerged in this research especially regarding other foreigners in Italy, a category called “with people from the birth country”, which had a typical frequency:

(…) I am very happy to share with competent people like GP008, like GP007, with whom we have a common vision, because the statute is shared and the common strategies too. We come from different backgrounds, but I discovered that, with the three of them, even free of charge, I achieved a lot with conviction, with passion […] with these two brothers (…).

##### With People from the Resettlement Country

The sense of community often was shown in the interviews towards the people from the resettlement country, which was found to have a varying frequency:

(…) I had everything from the community anyway because the operators who are there have always helped me, they have always given me advice that I have always followed, and that is the result I see today, practically everything starts from the community where I was (…).

#### 3.1.9. Positive Memories

Since the interview is also about exploring the refugee’s pre-migration situation, it sometimes brought out memories of the person from his birth country. Among these, two categories were identified. This domain was divided into two categories.

##### Of the Birth Country/Childhood

These conditions, differently from what one might think based on the sample, did not always present themselves as traumatic or depriving, but often they revealed pleasant childhood memories or the observation of positive aspects of one’s own culture. However, both were classified as variant:

(…) over there we live in harmony, we all live in the family, we all do things together, we never do things alone (…).

#### 3.1.10. Refusal

In some interviews, the refugee clearly stated the reason for departure from the home country. This domain was divided into two categories.

##### Of the Birth Country

Regardless of extreme situations such as wars or persecutions, some participants cited social and/or cultural conditions far from them and which they seemed to openly reject, in relation to the “birth country” (variant):

(…) There is practically not much solidarity. Then, I do not know, I don’t understand the Pakistani mentality, because I’ll explain it to you, Pakistan is not a poor country, it is not really a poor country, only that people do not pay taxes (…).

##### Past Events

The refusal was also mentioned in relation to “past experiences” that had concerned him (variant):

(…) I threw myself a bit into fundamentalism at a certain point, that is, for a few months, and there I really risked a lot, my beard just kept growing and my head shrunk (…).

## 4. Discussion

The aim of the present paper was to explore the development of place attachment toward the resettlement and origin country in a group of refugees with experiences of positive integration.

The first research question asked which factors determine the creation of a new place attachment. Through the frequency analysis, we found out that processes of identification, development of a sense of belonging, and an active commitment toward the resettlement and the origin community ease the construction of a new home.

In literature, there are no studies that describe how and if refugees create a new place attachment and a new identity and which factors help to determine it.

Consistent with the attachment theory [19,33], results from our research show that participants easily explored the resettlement country (both from a territorial and a community perspective) and established new relationships with natives. This data allows us to infer patterns of positive attachment that contrast with Ecke’s hypothesis that migrants are more likely to have insecure attachments than the non-migrant population [22].

In particular, the high frequency of the category “sense of belonging” aligns with Baumeister and Leary’s theory [34] that each of us has an innate motivation to belong, which the authors call a real need to belong. This need makes people committed to building and maintaining relationships with others, can motivate them to exhibit group-serving behaviours, and to focus more on the collective interest of the group in the social dilemmas of large groups [35]. The desire for interpersonal attachment is one of the most far-reaching and integrative constructs currently available for understanding human nature. It develops through frequent, emotionally pleasurable interactions with some people. When a social bond is broken, people seem to recover better if they form a new one. On the contrary, a lack of belonging, should constitute a serious deprivation and cause a series of negative effects.

Furthermore, a sense of belonging is strictly associated with identity, in particular with the social identity defined by Tajfel [36] as “the part of an individual’s self-concept which derives from his knowledge of his membership of a social group (or groups) together with the value and emotional significance attached to that membership” (p. 63).

As the author sustained [37], in order to define a “group”, external and internal criteria are required. The former is mostly related to the way the “outside” labels the group, while the latter refers to the processes of identification. This last process is made of two essential constituents: the awareness of being part of a group and the evaluation that a person makes of the same group. A third constituent is an emotional connotation related to the awareness and evaluation of social belonging. Social Identity is therefore part of a self-definition process that encompasses the person’s group belonging.

The category “identifying with both countries” is one of the most frequent categories (general), confirming Berry’s [38] hypothesis that integration does not mean losing one’s identity to conform completely to the new one, but being able to find a balance between them.

The second most common domain that can explain the process of building a new home was “expectations towards the resettlement country”. All its four sub-categories were classified as general, so they were present in almost all the interviews (they are “few opportunities”, “difficult life conditions”, “towards the birth country”, and “towards the resettlement country”).

Indeed, results show also that the construction of a new home is the result of an interplay among the expectations related to the resettlement country and participants’ active commitment toward its development and the development of the origin country. This peculiar result attests that participants recognized the need for an active effort in order to build their new home.

The second research question entered into the details of the dynamic between identity and the establishment of relationships with resettlement community members.

Firstly, some specific categories such as “desiring to learn the language” and “exchange culture” testify to some personal characteristics, such as an openness toward new experiences that may be at the base of the establishment of resourceful social relationships. Further studies should verify the presence of an association among refugees’ personological characteristics and their success in social integration.

It is interesting to note that the relationship between identity and the establishment of relationships with resettlement community members is characterized by a tension between seeking help from others and being a point of reference for others. Seeking help from others is a behaviour profoundly related to attachment. In particular, it refers to the dimension of affiliation defined by Murray as that of human beings to receive gratifications from harmonious relationships and a sense of community [39]. This is important for individuals who migrate, both towards people belonging to the birth country and both towards people coming from the resettlement country.

The motivation for affiliation has always been powerful and pervasive, and it motivates the way in which individuals form positive interpersonal relationships [34]. At the same time, caring for others seems to be closely related to affiliation and attachment. In fact, the ability of an individual to rely on or to place trust in the other depends on how (and if) caregivers demonstrated that people are trustworthy and served as a “secure base”. This is the origin of seeking help from others in difficult times (or, on the contrary, lack of trust in others), but also self-confidence. This attention and recognition allow an individual to grow up with an awareness of his or her own abilities.

Consistent with this, our participants show also to be a point of reference for others.

Indeed, after becoming acknowledged and attached to their new “home”, they put their own experience at service for others in order to ease their inclusion path.

As discussed in the results, in connection with existing literature about place attachment, refugees find their “home” in a place where they feel they belong, are well integrated, and are part of a community. In these people, “home” is not just the place where they are no longer persecuted, but a place where their identity has been reconstructed thanks to new positive relationships.

These patterns should be confirmed in future studies including wider and different refugee samples. These patterns would allow clinicians working with refugees not only to operate with positive integration, but a more complex understanding of the ways in which new place and interpersonal attachment develops.

### Limitations and Future Resarch

Although this type of research has allowed us to closely observe some important and delicate issues, we can affirm that it has some limitations.

Firstly, the participants came from Africa and South Asia (two-thirds of the interviewees come from Africa, three of them born in Afghanistan, one in Pakistan, and one in Kashmir), suggesting the need for a more representative geographic and cultural sample.

Another limitation is the language: some of the refugees did not speak Italian well and this meant that some words of the interviews were difficult to understand. In this way, sometimes the sense of what the refugee was saying was lost.

Another point to take into consideration is that qualitative research is particularly influenced by the researcher’s interpretation. This may have sometimes resulted in less accuracy and validity, although it is a strong point for grasping the complexity of the data collected. Nevertheless, teamwork is permitted to mitigate errors regarding the subjective interpretation, to ensure a fair degree of validity.

Additionally, only one study participant was female and future studies should make efforts to include a sample that more completely represents the population of individuals with protection status in Italy regarding gender, age, country of origin, religion, and other demographic variables. A larger and more representative sample would also permit more confidence in undertaking analysis of existing definitions of place attachment as was applied to our limited sample. Future studies could expand beyond refugee community association founders and leaders to include ordinary members of community associations as well as refugees who are not active in these contexts. Results would permit further reflection on Place Attachment, affiliation, and seeking help from others, and the role of refugees in supporting and advocating for their own communities.

Additionally, though the sample contained both refugees and asylum seekers, no distinction was made between the two groups in analysing interview data.

Although this work has provided much insight on place attachment among refugees, further research and analysis would be needed to investigate such complex issues as identity and attachment to the place.

## 5. Conclusions

This qualitative study explores the development of place attachment toward the resettlement and origin country in a group of refugees with positive integration experiences in the framework of attachment theory.

We tried to understand which factors determine the creation of a new “home” (RQ1), how new refugee identity develops, and how it relates to established relationships in the resettlement country (RQ2).

Regardless of the social sphere, it is difficult to find studies in the existing literature that specifically dealt with these issues in relation to refugees. For the most part, over the years, there has been a focus on the personological dimensions most frequently found in the migrant population. The issues of place attachment, affiliation, and seeking help from others are recurring and fundamental in the refugee population. They show that people who have experienced forced migration can construct a new home, a sense of belonging, and new positive relationships.

Regarding our hypotheses, analysis of interview data, in dialogue with existing literature, confirms the importance of place attachment whenever establishing in a new country. Attachment patterns influence and are influenced by the refugee tendency to trust other people and to seek help from others.

Additionally, we observed that place attachment in the resettlement country coexists with the attachment towards the birth country, showing that the same processes are involved.

This analysis leads us to conclude that in this selected sample, made up of successful refugees, there is an identity that does not deny its origins and roots, but which has now settled in a new place, Italy, and therefore feels that it belongs there. In fact, refugees did not manifest the will to leave or an intolerance in having arrived in Italy. They were instead motivated to stay, to help not only migrants who had lived their own experiences, but also Italian citizens. Consistent with recent contributions within the positive psychology branch of studies [25,26,40,41], our participants promoted the idea that it is possible to build a “plural society” in which diversity enriches and does not detract.

Furthermore, from a clinical perspective, the study of place attachment helps to better explore refugee’s adjustment, in order to identify what helps refugees to be an active part of a resettlement community. The comprehension of this phenomenon, and its promotion, could increase their integration and personal fulfilment.

This is a new field of research, and it needs further and more systematised studies.

## Figures and Tables

**Table 1 ijerph-18-08273-t001:** Participant Information.

Age	Gender	Nationality	Migratory Background	Years in Italy
M = 33.0 SD = 8.0	14 M 1F	3 Afghan 3 Somali 2 Malian 1 Algerian 1 Eritrean 1 Guinean 1 Kashmiri 1 Pakistani 1 Senegalese 1 Sudanese	10 Refugees 3 Asylum Seekers 2 Subsidiary Protection	M = 10.1 SD = 5.3

**Table 2 ijerph-18-08273-t002:** Results.

Domains	Categories	Subcategories	Frequencies
1. Sense of identity	Acquiring the identity of the resettlement country	Desiring to learn the language	General
		Adapting to the culture	Typical
	Maintaining the identity of the birth country	Cultural and linguistic adaptation	Variant
		Difficulty in integrating	Variant
	Having an identity with both countries		General
2. Expectations towards the resettlement country	Derived from difficulties from birth country	Few opportunities	General
		Difficult life conditions	General
	Derived from a sense of responsibility	Towards the birth country	General
		Towards the resettlement country	General
3. Sense of belonging	Feeling at home		Typical
4. Community integration	Inclusion in the italian community	Complex	Typical
		Simple	Typical
	Inclusion in the foreigner’s community	Complex	Variant
	Non inclusion	In the resettlement community	Typical
		In the foreigner’s community	Typical
5. Trust	Seeking help from others	In the resettlement country	Variant
		In the birth country	Typical
	Self-confidence		Variant
	Difficulty in trusting		Variant
6. Opportunity seizing	Exchange culture		Typical
7. Being a point of reference for others	For the birth country		Typical
	For the resettlement country		Typical
8. Sense of community	With people of the birth country		Typical
	With people of the resettlement country		Variant
9. Positive memories	Of the birth country		Variant
	Of childhood		Variant
10. Refusal	Of the birth country		Variant
	Past events		Variant

## Data Availability

The data are not available due to ethical concerns in an effort to protect the privacy of the vulnerable population at the center of the study.

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
