# Peer review of "At Home: Place Attachment and Identity in an Italian Refugee Sample"

_ijerph, 2021, doi:10.3390/ijerph18168273_

Round 1

Reviewer 1 Report

I read the manuscript entitled "At Home: Place Attachment and Identity in an Italian Refugee Sample" and found it very interesting and well-written. I have made some few annotations.

Author Response

Response to Reviewer 1 Comments

Point 1: I am not familiar with the expression “mother earth”, except to refer to mother earth which cannot be intended meaning here. Do you mean something like native country, or homeland?

Response 1: Thank you for your suggestion. We decided to renominate this word as “homeland”.

Point 2: Delete “tie”.

Response 2: Unfortunately, we could not delete this word since it gives meaning to the phrase.

Point 3: Delete “exploit”.

Response 3: We deleted the word.

Point 4: There is a contradiction. Either the process is one-dimensional, or it makes sense to identify different dimensions.

Response 4: Thank you for your annotation. The use of the term “one-dimensional” was inappropriate, so we decided to delete it.

Reviewer 2 Report

Thank you for the opportunity to review the paper “At Home: Place Attachment and Identity in an Italian Refugee Sample”.

Overall, the topic of this paper is very interesting and worth of wide discussion on the importance of place attachment for migrants/ refugees.

In addition, the paper is very well organized and documented.

However, there are a few concerns which need to be addressed.

  • I suggest the authors to separate more clearly the interviewees’ responses from the text. For example, the authors could use the italic font for the responses or blank spaces between text and responses.
  • I suggest the authors to rename the section 1. Limitations as 4.1. Limitations and future research since it included recommendations for future studies.

I wish the authors all the best!

Author Response

Response to Reviewer 2 Comments

Point 1: I suggest the authors to separate more clearly the interviewees’ responses from the text. For example, the authors could use the italic font for the responses or blank spaces between text and responses.

Response 1: Thank you for your suggestion. We used blank spaces between text and responses.

Point 2: I suggest the authors to rename the section 1. Limitations as 4.1. Limitations and future research since it included recommendations for future studies.

Response 2: Thank you for your suggestion. We renamed this section 4.1 Limitations and future research.
